# Comparative Analysis of Tick-Borne Relapsing Fever Spirochaetes from Ethiopia and Nigeria

**DOI:** 10.3390/pathogens12010081

**Published:** 2023-01-03

**Authors:** Adefolake A. Bankole, Bersissa Kumsa, Gezahegne Mamo, Ndudim I. Ogo, Nusirat Elelu, Winston Morgan, Sally J. Cutler

**Affiliations:** 1Department of Bioscience, School of Health, Sport & Bioscience, University of East London, London E16 2RD, UK; 2Department of Pathology and Parasitology, College of Veterinary Medicine & Agriculture, Addis Ababa University, Bishoftu, Addis Ababa P.O. Box 34, Ethiopia; 3Department of Veterinary Microbiology, Immunology, and Public Health, College of Veterinary Medicine & Agriculture, Addis Ababa University, Bishoftu, Addis Ababa P.O. Box 34, Ethiopia; 4Livestock Investigation Division, National Veterinary Research Institute, Vom 930010, Nigeria; 5Department of Veterinary, Public Health and Preventive Medicine, Faculty of Veterinary Medicine, University of Ilorin, Ilorin 240003, Nigeria

**Keywords:** soft ticks, ruminants, Ethiopia, Nigeria, tick-borne relapsing fever

## Abstract

Despite increasing reports of tick-borne diseases in Africa, remarkably, reports of tick-borne relapsing fever (TBRF) in Nigeria are lacking. *Ornithodoros savignyi* from Nigeria have been reported with the relapsing fever *Candidatus* Borrelia kalaharica. Conversely, in Ethiopia, the agent of relapsing fever is the louse-borne relapsing fever (LBRF) spirochaete *Borrelia recurrentis* with no TBRF reported to occur. A total of 389 *Ornithodoros* ticks, Ethiopia (N = 312) and Nigeria (N = 77), were sampled, together with 350 cattle, and 200 goat sera were collected from Nigeria. Samples were screened for *Borrelia* spp. by RT-PCR. Reactive samples were confirmed, then sequenced using flagellin B, 16S rRNA, and 16S–23S intergenic spacer region. The prevalence of *Borrelia* spp. in livestock was 3.8% (21/550) and 14% (3/21) after final molecular confirmation. Of 312 ticks from Ethiopia, 3.5% (11/312) were positive for *Borrelia*, with 36% (4/11) by conventional PCR. Sequencing revealed that the borreliae in soft ticks was *C.* B. kalaharica, whilst that found in animals was *Borrelia theileri*. Soft ticks were confirmed by sequencing 7% (22/312) and 12% (9/77) of the Ethiopian and Nigerian ticks, respectively. Phylogenetic analysis revealed that these were *Ornithodoros savignyi*. This is the first evidence of *C*. B. kalaharica in Ethiopia and demonstrates the co-existence of TBRF in a country endemic to LBRF. Important, this might cause a diagnostic challenge given that LBRF is predominantly diagnosed by microscopy, which cannot differentiate these two spirochaetes. Furthermore, we report *B. theileri* in ruminants in Nigeria, which may also be of veterinary and economic importance.

## 1. Introduction

Most tick-borne relapsing fever (TBRF) are zoonotically caused by the bacteria of the genus Borrelia that are majorly vectored by soft *Ornithodoros* ticks. The existing dogma is that these ticks inhabit the burrows of their preferred vertebrate hosts, such as rodents and hedgehogs, and feed indiscriminately upon them [1] serving to ensure the circulation and maintenance of these spirochaetes within the host-tick ecosystem. Many can also undergo transovarial transmission, but this is not believed to be sustainable over multiple generations [2]. However, it can provide a valuable demonstration of tick vector competence [2]. These ticks have secondarily adapted to domestic animal and human dwellings, often taking advantage of anthropogenic factors such as densely populated clusters of people and livestock [3].

Clinically, relapsing fever (RF) is characterized by several episodes of recurrent fever, usually lasting between 2 to 7 days, interspersed with asymptomatic periods of up to 10 days [4]. This is often accompanied by a wide range of symptoms such as headache, nausea, myalgia, vomiting, etc., and in severe cases of neurological complications such as meningoencephalitis, eye defects, hepato-nephritis, and sudden abortion in women can occur [5]. A Jarisch–Herxheimer reaction can occur upon antibiotic treatment, again necessitating clinical management [6]. In animals such as livestock and poultry, TBRF, such as bovine borreliosis, is transmitted by hard-bodied ticks mostly belonging to the genus *Rhipicephalus* spp. [7], whereas the soft tick *Argas persicus* transmits avian spirochaetosis [8]. Avian spirochaetosis is caused by *B. anserina*, which can be severe for poultry and economically damaging, with death reported approximately 12 hours after the onset of symptoms during the acute phase and 6 to 8 days for milder cases [9]. In contrast, *B. theileri*, the agent of bovine borreliosis, exhibits mild symptoms of fever, lethargy, and anemia when compared to other pathogenic agents such as *Babesia* spp. [10], *Theileria* spp. [11], and *Trypanosoma* spp. [12], which exhibits similar clinical symptoms but in a more aggressive form. Notwithstanding, bovine borreliosis results in reduced animal productivity, which can have a negative economic impact [13]. Historically, livestock borreliosis have been reported in some African countries such as Ghana [14], Nigeria [15], and Mauritius [16] using blood smear microscopy, and more recently in Zambia where the spirochaete was detected in wild and domestic animals using molecular tools [13]. A study from Nigeria using a controlled lab setting was able to infect calves with the prodigy of an infected female *Rhipicephalus* (formerly *Boophilus*) *annulatus* tick previously collected from cattle. The spirochaetes were detected in blood smears 18 days after initial exposure to the ticks; however, molecular confirmation was not performed. Notwithstanding, this was one of the studies that provided substantial evidence that *R. annulatus* is a competent vector for *B. theileri* [15].

In East Africa, the predominant TBRF-causing spirochaete is *B. duttonii*, exclusively vectored by *O. moubata* [17]. The incidence rate in some regions, such as Tanzania, has been as high as 384/1000 among children of less than one year, and the perinatal death rate of 436/1000 births over a 5-year period [18]. In addition, louse-borne relapsing (LBRF) remains endemic within some regions of East Africa, particularly Ethiopia, where the mortality rate can be as high as 40% in untreated cases and 5% in treated patients [19]. Although the presence of TBRF in Ethiopia has been queried, no evidence of its presence has been forthcoming to date. Some TBRF borreliae have been described from Ethiopia, such as *B. anserina* from Ethiopian *Argas persicus* ticks [20] and an RF *Borrelia* spp. in *Amblyomma cohaerens* ticks [21]. In West Africa, TBRF is predominantly caused by *B. crocidurae* with *O. sonrai* as its vector [22]. Most cases of the diseases are found in the rural areas of Saharan, Sudano-sahlian, and Sahelian countries [22], where they account for around 12% of febrile illness in all age groups, and up to 16–18% in children between 8–14 years and adults between 22–24 years respectively [23]. In North Africa, the agent of TBRF is *B. hispanica*, vectored by *O. erraticus* ticks [24]. In Nigeria, there is little knowledge or clinical suspicion of TBRF existing within the country, though the presence of *C*. B. kalaharica and its vector *O. savignyi* was recently described [25]. 

Given the presence of a proposed agent of TBRF in Nigeria *C*. B. kalaharica, and the possibility of TBRF existing in Ethiopia, the aim of the study was to determine the presence and clinical or veterinary relevance of TBRF in both countries.

## 2. Materials and Methods

### 2.1. Nigeria

In 2018, soft ticks were obtained from the laboratory of the National Veterinary Research Institute, Vom. Fifty soft ticks were previously collected from cattle farms in Vom area of Jos South local government area (LGA) 9.7376° N/8.8087° E in Plateau State, Nigeria with an estimated temperature between 21 and 25 °C [26]. An additional 27 soft ticks previously collected around camel shelters in Maiduguri LGA 11.846920° N/13.157120° E in Borno State (Figure 1) [27], Nigeria, where the average temperature is 37 °C with an estimated annual rainfall of 600–1200 mm [28].

### 2.2. Ethiopia

Soft ticks were manually collected from cattle shelters in Amibara district in zone 3 Eastern Afar Regional State in Ethiopia (Figure 1) in 2020. The ticks were mainly concentrated in the dusty soil where these animals rest (Figure 2). Amibara district is 250 km away to the northeast of Addis Ababa, with an estimated elevation level between 550 and 1100 m above sea level, an average temperature of 29 °C, and a yearly precipitation level between 400–600 mm. The state is comprised of 18 pastoral associations (PAs), which is the largest in Africa [29].

### 2.3. Animal Study

In order to determine the presence of *Borrelia* in livestock, 2 mL blood samples were collected via jugular vein puncture from cattle and goat farms between April and May 2019 in Borno and Plateau States, Nigeria (Figure 2). Samples were left to clot, and the serum was transferred into plain sterile tubes and stored at −20 °C prior to transport and analysis in the UK under the DEFRA license (ITIMP19.0310).

One hundred and forty-five cattle sera were collected from Jere LGA (11.84690° N–13.157120° E; elevation 1060 meters above sea level) in Borno state (Figure 1).

Two-hundred-and-five cattle sera were collected from Bokkos (9°19′024″ N–8°55′197″ E; 9°19′078″ N–8°54.891′ E) and Barkin-Ladi (9°31′334″ N–8°49′014″ E; 9°26′292″ N–8°56′289″ E). Two-hundred goats were sampled locally from house to house in Jos-South (9°32′00″ N–8°54′00″ E), Shendam (8.8955° N, 9.4537° E), and Kanke (9.3702° N, 9.5896° E) LGAs of Plateau state, with an elevation level of 4000 meters above sea level (Figure 1).

### 2.4. DNA Extraction and PCR Amplification

Prior to extraction, each tick was rinsed in phosphate-buffered saline (PBS), air dried, and halved into two equal parts using individual sterile scalpels. Halved ticks were placed in a 2 mL screw-capped tubes containing homogenization lysing beads (Matrix H) (MP biomedicals) and 200 μL PBS before crushing at a speed of 6.5 m/s for 5 min, depending on the tick size, using the FastPrep-24^®^ 5G (MP biomedicals) tissue homogenizer. Total DNA was extracted using DNeasy blood and tissue kit (Qiagen^®^) according to manufacturer’s guidelines (Qiagen^®^, Hilden, Germany). Homogenized ticks were resuspended in proteinase K and lysis buffer at a volume of 1:10 (20 μL in 180 μL for the bigger ticks and 10 μL in 90 μL for smaller ticks) respectively for each sample. Ticks were incubated overnight in the water bath at 56 °C as previously described [25,30].

For livestock sera, 100 μL of each sample was digested in 20 μL proteinase K and the final volume adjusted to 220 μL with phosphate-buffered saline (PBS). Sera samples were incubated for one hour, as stated above. Lysates of both tick and sera samples were treated with sodium buffer, vortexed, and incubated in the water bath at 56 °C for 10 min. Absolute ethanol (100%) was added to each sample and vortexed to precipitate the DNA from the solution. Samples were transferred into individual spin filter columns and rinsed with buffer containing alcohol to remove cellular debris and other contaminants. A final volume of 200 μL of each extracted DNA was eluted, and samples frozen at −20 °C pending investigation. 

A 450 bp tick amplicon was produced using 16S rRNA primers [31] (Table 1). All conventional PCRs were run using the Bio-Rad^®^ T100™ Thermal cycler at a final volume of 25 μL and with the following cycling conditions; denaturing at 94 °C for 5 min, followed by 35 cycles of 94 °C for 45 s, annealing of 50 °C for 45 s, 72 °C for 45 s, and a final extension of 72 °C for 10 min. DNA of an *Ixodes ricinus* and *Ornithodoros savignyi* tick were used as the positive template, while nuclease-free water was used as the negative template control. Amplified PCR products were viewed using 1.5% agarose gel electrophoresis stained with SYBR safe (Invitrogen^®^) using a 100 bp DNA ladder to determine the sizes of the amplicons (either Invitrogen^®^; or New England BioLabs^®^) at 100 volts for 80 min.

Extracted ticks and livestock DNA were screened for the presence of *Borrelia* by using *Borrelia*-specific real-time polymerase chain reaction (qPCR) primers (Merck) designed to amplify a 148 bp fragment of the 16S rRNA of *Borrelia* as previously described [32]. A final PCR volume of 25 μL containing reaction buffer (1X), dNTPs (0.2 mM each), MgCl_2_ 2.5 mM), 2 μL DNA template, each primer at (500 mM), probe (250 mM), and 0.15U taq polymerase. Cycling conditions were as follows: 95 °C for 10 min, 40 cycles of 95 °C for 15 s, and 60 °C for 1 min as previously described [25]. DNA of *B. burgdoferi senso stricto* B31 strain was used as positive control and nuclease-free water (Invitrogen) as negative control. Amplification was conducted using the Agilent AriaMx*®* 1.2 Real-Time thermocyclers. Samples with a cycle threshold (CT) value of ≤36 were considered positive. All *Borrelia*-positive tick samples from the initial qPCR screening were retested and subjected to further confirmation of the flagellin B (flaB) gene and the 16S–23S intergenic spacer region (IGS) by nested PCR. *Borrelia*-positive animal samples were confirmed using the 16S rRNA and flaB nested PCRs with the primers listed in (Table 1).

Cycling conditions for tick *Borrelia* DNA; FlaB: 95 °C for 5 min, 40 cycles of 95 °C for 30 s; 56 °C for 30 s; 72 °C for 90 s, and a final extension of 72 °C for 10 min [33]. IGS first round: 94 °C for 3 min, 35 cycles of 94 °C for 30 s; 56 °C for 30 s; 72 °C for 60 s, and a final extension of 72 °C for 7 min; nested second round: 94 °C for 3 min, 35 cycles of 94 °C for 30 s; 60 °C for 30 s; 72 °C for 60 s, and a final extension of 72 °C for 7 min [17]. 

Cycling conditions for livestock *Borrelia* samples; 16S rRNA first round: 94 °C for 4 min, 35 cycles of 94 °C for 30 s, 50 °C for 30 s, 72 °C for 45 s and a final extension of 72 °C for 10 min; and for the nested second round: 94 °C for 4 min, followed by 30 cycles of 94 °C for 30 s, 60.2 °C for 30 s, 72 °C for 35 s, and a final extension of 72 °C for 10 min [35]. FlaB first round; 30 cycles of 94 °C for 40 s, 63 °C for 60 s, 72 °C for 60 s; nested second round: 30 cycles of 94 °C for 40 s, 50 °C for 40 s, 72 °C for 40 s [34].

Amplicons were evaluated using 1.5% agarose gel electrophoresis stained with SYBR safe (Invitrogen^®^). Borrelial amplicons successfully generated were purified using the Qiaquick^®^ PCR purification kit (Qiagen) according to manufacturer’s protocols, DNA concentration was measured using the Nanodrop ND-1000 spectrophotometer (Thermo Scientific™, Wilmington, NC, USA) before Sanger sequencing.

### 2.5. Phylogenetic Analysis

The prevalence of *Borrelia* infection in soft ticks and animals was determined using descriptive statistics. Statistical significance was set at *p* < 0.05. Resulting sequences derived from this study were compared to those in the GenBank^®^ using BLASTn (www.ncbi.nlm.nih.gov/BLAST, accessed on 3 August 2022). Multiple sequence alignments were performed using the MUSCLE program and phylogenetic construction using the neighbor-joining and Maximum Likelihood methods at a confidence test of 1000 bootstrap on MEGA 11 [38].

Sequences generated from this study have been deposited into the GenBank under the accession numbers (OP688109, OP688110, OP688111, OP688112, OP688113, OP688114, OP688115, OP688116, OP688117, OP745098, OP745099, OP745100, OP745101, OP703387, P703388, OP839113, OP839114, OP839115). Raw sequences can be found in the Appendix A.

## 3. Results

### 3.1. Tick Identification

A total of 22 (7%) of the 312 and 20 (26%) of the 77 Ethiopian and Nigerian soft ticks, respectively, yielded positive amplicons of 450 bp on standard PCR using tick 16S primers listed in Table 1. Of these, a random convenience subset of ten ticks from each country was submitted for Sanger sequencing. 

### 3.2. Sequence Analysis for Tick Identification

Using the 16S rRNA tick-specific gene target, sequence divergences between the Ethiopian and Nigerian tick sequences were observed, as well as a few heterogeneities among ticks from the same location (Figure 3). Ethiopian tick samples SC4, SC2, SC8, SC9, and SC80 (listed from numbers one to six in Figure 3) at 450 bp had a (94.15%, 94.7%, 93.89%, and 94.4%) similarity to *Ornithodoros savignyi* KJ133578, MF415646 from Sudan [32,33] and KU163242 from Egypt [32]. Representative sequences have been deposited into GenBank under the accession numbers OP688111, OP688113, OP688114, OP688116, and OP688117. Single nucleotide polymorphisms (SNPs) were observed at positions (250–251 bp; 356–357 bp) in sample SC8 (Figure 3). Notwithstanding these slight polymorphisms, the sequences clustered closely together within the phylogenetic tree (Figure 4). For the Nigerian ticks, samples SCP24 and SCP16 (listed from numbers 7 to 10 in Figure 3) from Plateau state had similar sequences with a score of 98.11% and 96.28%, and Borno samples SCB1, SCB3 had similarities of 98.45% and 99.09% respectively to *O. savignyi* from Egypt (Figure 3). SNPs were detected at (31–32 bp, 59 bp, 96 bp, 100 bp, 115 bp, 132 bp, 137–138 bp, 162 bp, 196 bp, 338 bp, 417–418 bp, and 474 bp), most of which were observed in sample SCB3 as evident on the tree (Figure 4). All four sequences have been deposited into the GenBank under accession numbers OP688109, OP688110, OP688112, and OP688115.

When comparing the tick sequences between the two countries, several differences highlighted with parallel vertical lines were observed at positions 86 bp, 105 bp, 148 bp, 151 bp, 180 bp, 182 bp, 210 bp, 213–214 bp, 223 bp, 225–226 bp, 234 bp, 272 bp, 276 bp, 287 bp, and 300 bp (Figure 3). These significant differences resulted in a divide between the Nigerian and Ethiopian ticks within the phylogenetic tree (Figure 4).

### 3.3. Polymerase Chain Reactions for Borrelia Infections

Eleven (3.5%) out of the 312 ticks from Afar Regional State in Ethiopia were infected with *Borrelia* upon RT-PCR, whilst none of the 77 ticks from Nigeria harbored the spirochaete (Table 2). Further confirmation of the positive tick samples using borrelial flagellin and the 16S and 16S–23S intergenic spacer region (IGS) for species characterization yielded amplicon sizes of 770 bp and 750 bp respectively, for 4 of the 11 screen positives Ethiopian ticks. There was no significant difference in the level of infection between the adult ticks or nymphs or if ticks were engorged versus non-engorged (*p* > 0.05) (Table 3).

Alongside this, 18 (5%) of the 350 cattle sampled from different herds in Borno state where *C*. B. kalaharica had been previously reported [25] and Plateau state, Nigeria tested positive for borrelial infection using RT-PCR, while 3 (1.5%) of the 200 goats from Shendam and Kanke areas of Plateau state tested positive by RT-PCR (Table 2). After conventional PCR confirmation, the three goat DNA yielded positive amplicons of 323 bp and 622 bp for both the flagellin and 16S rDNA genes, respectively.

### 3.4. Sequence Analysis for Borrelia

Analysis of the four Ethiopian borrelial flagellin sequences by BLAST revealed that all but sample (SC7, 92.24%) showed a >99% to 100% identity with *C*. B. kalaharica strain that has also been reported from Nigeria [25]. Among these 770 bp sequences, there were some divergences, particularly with sample SC7 that had the most single nucleotide polymorphism (SNPs) compared to the other three samples, SC4, SC56, and SC2 (Figure 5). Alongside SC7, three nucleotide differences were also observed in sample SC2, both of which were evident in the phylogenetic tree (Figure 6). Representative sequences were deposited into the GenBank under accession numbers (OP839113 and OP839114).

For the Ethiopia samples, borrelial IGS sequences of 750 bp were obtained for all four tick extracts. These were homologous and showed >99% similarity with the *C*. B. kalaharica Nigerian strain (Figure 7). Sequences were deposited in the GenBank under accession numbers OP745098, OP745099, OP745100, and OP745101.

*Borrelia* 16S rRNA sequences from Ethiopian ticks were excluded from the study as they were of poor quality, which made it impossible to use them for phylogenetic comparison. 

Borrelial flagellin sequences of 350 bp from Nigerian goat samples CS51, CS19, and KAD8 were highly related in their sequences, revealing similarities of 97.81%, 97.73%, and 98.89% similarity respectively, to *B. theileri* ON191583 from Brazil [39]. These sequences also aligned closely to a strain recently reported from cattle and impalas from Zambia [13], Brazil [40], Argentina [41], and Mali [42] (Figure 8). Interestingly, a sequence reported as *B. theileri* from Israel did not align with any of the *B. theileri* sequences on the tree but instead aligned next to *B. lonestari* from the USA and formed a sister clade with *B. persica* from Israel. A representative sequence was deposited in the GenBank under accession number (OP839115).

Two of the three initial screen-positive borrelial goat sera (CS51 and CS19) produced 650 bp sequences by conventional 16S nested PCR. BLAST analysis of CS51 and CS19 revealed a 99.83% and 99.67% identity, respectively, to *B. theileri* LC656246 [13]. These two *Borrelia* goat sequences were deposited in the GenBank under accession numbers OP703387 and OP703388. The sequences aligned closely with those reported from Colombia [43], Thailand [44], and Zambia [13]. The 16S rRNA gene is known for its poor discriminatory power. Hence, it is not very useful for phylogenetic information, as shown in (Figure 9).

### 3.5. Phylogenetic Construction

Phylogenetic trees were created to gather information on the genetic correlation of our resulting sequences with those of other *Borrelia* spp. and *Ornithodoros* spp. in the GenBank. Using flab, *Borrelia* detected from Ethiopia aligned closely with the clade of *C*. B. kalaharica previously reported in *O. savignyi* ticks from Nigeria (Figure 6) [25]. The IGS *Borrelia* sequences similarly aligned with the *C*. B. kalaharica reported from *O. savignyi* ticks from Nigeria, as well as a novel *Borrelia* spp. reported in humans from Tanzania [17]. Conversely, the flagellin and 16S rRNA *Borrelia* sequences from the Nigerian goats showed close similarity with the *B. theileri* reported in cattle from Zambia [13] and Columbia [43], as well as in Ixodid ticks from Thailand [44] (Figure 8 and Figure 9). 

## 4. Discussion

We investigated the presence of *Borrelia* spirochaetes in soft ticks from Ethiopia and Nigeria and in ruminants from Nigeria. Molecular tools were used to characterize these to their species level. The *Borrelia* sequences of the flagellin and intergenic spacer region (IGS) from Ethiopian ticks revealed a 99–100% identity to the proposed ‘*Candidatus* Borrelia kalaharica’ from Nigerian ticks and those from human tourists returning home from other locations in Africa with clinical relapsing fever [45,46]. Although a few polymorphisms were detected in some of the sequences, they did not impact the phylogenetic information provided by these molecular alignments. 

None of the ticks from Nigeria were positive for *Borrelia*. This was surprising given that some of these ticks were collected from a region where *C.* B. kalaharica had previously been reported [25]. The original report worked with pooled samples, whereas in this study, individual ticks were evaluated, which could have accounted for this finding. Some of the Nigerian ticks used in this study were collected around camel shelters rather than cattle markets. It is possible that camels might not be competent hosts for this spirochaete, as demonstrated for *Borrelia burgdorferi* sensu lato and deer [47,48], where certain host species clear spirochaetal carriage through host incompatibility. In contrast, the ticks collected from cattle farms are from a region where neither *Borrelia* nor soft ticks have been previously reported. Our failure to detect positive Nigerian ticks might also reflect the smaller sample size of 77 as opposed to the 312 ticks evaluated from Ethiopia or the possible focal occurrence of this spirochaete within ticks. This finding is unlikely to have risen from methodological differences as the same protocol was used for all ticks and our controls showed the assays were performing as expected. Certainly, these finding warrants a further extensive investigation.

Whether TBRF might co-exist alongside the endemic louse-borne relapsing fever (LBRF) in Ethiopia has been queried for some time [20]. Indeed, in the 1930s, attempts to infect *O. savignyi* with *B. recurrentis*, the agent of LBRF, were unsuccessful [49]. We have demonstrated the presence of *C.* B. kalaharica in *O. savignyi* ticks collected from Ethiopia, establishing that spirochaetes capable of causing TBRF do exist in this country. Whether these have been causing clinical or veterinary health impacts remains to be determined. With microscopy being the predominant method of diagnosis in developing countries which does not differentiate between these different spirochaetes, there is a strong possibility for misdiagnosis [19]. 

The human significance for *C*. B. kalaharica is obviously a key question. Interestingly, relapsing fever resulting from the *O. savignyi* tick was reported in Somalia back in 1936 [50]. These cases showed seasonality from April to September when the population was fewer in numbers, with the disease being mild with only six deaths reported amongst the 981 treated cases. As that report did not exclude the possibility of potentially co-existing with LBRF, the disease could not be directly attributed to the bite of *O. savignyi* ticks. More conclusive evidence was provided by a case reported in 2016 by a German tourist that visited the Kalahari Desert of Botswana and Namibia in Southern Africa, an area endemic to soft ticks [51], who returned home with a fever and was found to have borreliae present in her Giemsa-stained blood film [45]. This was followed by a further report of another traveler who also visited South Africa, Botswana, Namibia, and Zimbabwe, returning with recurring fever episodes [46]. Infection in humans has clearly been described, but the clinical impact on those with greater exposure to these ticks remains to be assessed. Interestingly, the IGS sequences of *C*. B. kalaharica show considerable similarity to those described from another poorly described borrelial species present in Tanzania [52,53,54]. This borrelial species was not only found in ticks but was observed in the blood of children residing in the same village with and without fever [52]. The name *B. mvumii* was tentatively suggested for this spirochaete, but the criteria for species status have not yet been fulfilled [55]. Whether *O. savignyi* ticks co-exist alongside the endemic *O. moubata* ticks in this region remains to be ascertained. Morphologically to the untrained eye, these tick species look similar. However, they can be differentiated by the presence of eyes in *O. savignyi* that are absent in *O. moubata*.

The Ethiopian ticks were 93–94% identical to ‘*Ornithodoros savignyi*’, the sand tampon from Sudan, using the 16S rDNA gene. Historically, this tick has been previously reported in neighboring Eritrea [56] and is suspected to be in Ethiopia but was not confirmed until this study [50]. Furthermore, *O. savignyi* is endemic in Sudan and Somaliland, both of which share land borders with Ethiopia. The latest taxonomic revision of the Afrotropical *Ornithodoros* species has described variations between *O. savignyi* from Sudan to those in Somaliland and Egypt based on morphological characteristics. The lower homology of the Ethiopian ticks to *O. savignyi* sensu stricto could also reflect these differences. The reclassification also suggests the possibility of a Middle Eastern species representing a distinct species compared to the North African ticks based on the genetic differences observed in their 16S sequences [51]. In contrast, the Nigerian ticks showed high homology to *O. savignyi* from North Africa (Figure 4), confirming that differences in ticks from Nigeria and Ethiopia may be based on genetic and geographical factors.

All Ethiopia tick sequences were highly related, apart from a few polymorphisms. The *O. savignyi sensu lato* complex has recently undergone taxonomic revisions. *O. savignyi* from North Africa probably has the widest geographical distribution facilitated by dispersal along camel trading routes through to the Middle East. Other members of the *O. savignyi* complex have been reclassified into *O. noorsveldensis* for those in the Eastern Cape and *O. pavimentosus* for those from the Kalahari region [51,57]. Comprehensive tick surveys that have mapped the presence of relapsing fever borreliae and their tick vectors may have overlooked the significance of *O. savignyi* complex ticks. These studies tended to follow the conventional zoonotic paradigm that ticks are primarily feeding on rodents and residing in their burrows, with human infection being more accidental following the non-specific feeding of these ticks [1,22]. Conversely, *O. savignyi* gravitates where livestock congregates, with these animals serving as a major blood meal for all life stages of the tick. In developing countries, livestock is often housed in proximity to humans [58] which is a typical case in all the study sites where livestock live in close contact with humans, thus presenting the considerable zoonotic potential for infection with tick-borne pathogens.

The overall *Borrelia* prevalence of 1.3% in *O. savignyi* ticks from this study is lower than the 6.1% reported in pooled ticks from Nigeria [25] but comparable to 2.6% reported in soft ticks from Israel [59]. It is difficult to determine the precise level of infectivity from the earlier study among the pooled ticks. Of the 312 Ethiopian ticks screened, only 7% of the ticks produced amplifiable products using the 16S gene target. Whether this low level of amplification is caused by sequence heterogeneity remains to be explored. However, the DNA of one *Ixodes ricinus* and one *O. savignyi* tick were used as positive templates, and nuclease-free water was used as the negative template control, demonstrating that our assay worked as expected. Failure to reliably amplify *O. savignyi* tick 16S rRNA has been previously reported [25]. It is unlikely that this low level of amplification of tick 16S rRNA resulted from DNA extraction problems, as nanodrop data suggested adequate DNA yields. Furthermore, *Borrelia* was detected in a sample that failed to produce tick 16S rRNA, supporting the notion that this was not a result of poor extraction or the presence of PCR inhibitors. In addition, ticks were sampled by third parties from Ethiopia and Nigeria pre-pandemic; it is possible that the lengthy storage of samples before analysis post-pandemic affected the integrity of the samples. These samples were stored in ethanol, and prolonged storage has not prevented recovery of *Borrelia* from museum tick samples stored for considerably longer than the samples we describe herein [60].

Sera from livestock from two states in Nigeria; Borno state, an area endemic to soft ticks [22], and Plateau state, an area not known to be endemic for soft ticks until this study, were tested for the presence of borreliae by PCR. The *Borrelia* detected in the goat from Nigeria was *B. theileri*, the causative agent of bovine borreliosis. This borrelial species has not been previously reported in animals in Nigeria using molecular tools. Historically, a study in the 1970s in Plateau state demonstrated that ticks from the genus *Rhipicephalus* (formerly *Boophilus*) are competent vectors for *B. theileri* after successful transmission of spirochaetes to a calf. Subsequently, some of the progeny of the engorged ticks that fed on the first calf was attached to a second calf and were also able to infect the animal [15]. 

The overall *Borrelia* prevalence rate of 1.5% (3/200) *Borrelia* in livestock is lower than the 3.3% (42/1260) reported in cattle from neighboring Cameroon [12]; 4.1% (20/488) from Zambia [13]; 17.9% (40/225) in Botswana [7], but higher than the 0.5% reported in *Rhipicephalus geigyi* ticks from Mali [42]. The evidence suggests that the level of borreliosis in livestock across Africa is relatively varied. This borrelial infection does not appear to be zoonotic but could impact the economic productivity of livestock. Interestingly, there has been a report of *B. theileri* detected in a human head louse [61], but whether this represented passive spill-over or is capable of onward transmission remains an open question.

Hard ticks of the genus *Rhipicephalus* spp. are the established vectors of *B. theileri* [15,40], and these species are of huge veterinary burden in Nigeria as they parasitize both livestock and domestic animals [30,58]. These ticks are known to hide in vegetation where they wait for a suitable host [62]. The roaming and foraging nature of goats makes them an ideal host for ticks for their blood meal. Relapsing fever borreliosis in livestock causes economic losses, particularly for individuals and countries that are largely dependent on livestock farming and trade [7]. 

Interestingly, the *B. theileri* from Israel KP191622 did not align next to any of the *B. theileri* flagellin sequences on the phylogenetic tree (Figure 8); instead, it aligned next to *B. lonestari* from the US and formed a sister clade next to *B. persica* reported in *O. tholozani* ticks and humans in Israel [59]. Whether this report represents an initial misidentification remains to be ascertained. Notwithstanding, there is another report of *B. theileri* being detected in *O. tholozani* ticks collected from caves in Israel by the same researchers [63]. This tick is the natural vector of *B. persica*, another agent of TBRF in Central and Middle Asia, India, and Egypt [64,65]. Whether these ticks were competent vectors to give an onward transmission of *B. theileri* remains to be established, but all positive ticks had previously been fed upon cattle [63], which might have been an accidental spill-over infection. *O. tholozani* is known to be multifaceted in harboring different pathogens such as *Coxiella burnetti* [66] and the West Nile virus, which was suspected of having infected birds in the former USSR [67]. Whether *O. tholozani* ticks could serve as competent vectors to give an onward transmission of *B. theileri* remains to be established [61].

The identity of ticks collected from Nigeria were confirmed as *O. savignyi*, with sequence heterogeneity observed between the ticks from Borno and Plateau states. These divergences could have risen because of geographical factors such as climate and altitude. For instance, Plateau state is more than 4000 feet above sea level and enjoys more temperate weather with an average monthly temperature of 21 °C to 25 °C compared to the rest of the nation [26], whereas Borno is one of the hottest states in the country where drought is endemic due to short spell of rainfall that is followed by a long dry spell, with an average annual temperature of 37 °C [30] 

This geographical difference may play a role in nucleotide divergences. However, the impact of the climate on soft ticks is poorly understood partly due to their enclosed lifestyle of living in proximity of the burrows of their small mammal host, which makes them more difficult to find in nature compared to hard ticks that quest for a longer period in open spaces [68]. 

## 5. Conclusions

In conclusion, this study reports the presence of ‘*Candidatus* Borrelia kalaharica’ a human relapsing fever agent in soft ticks in the Eastern Afar Region in Ethiopia. In this region, it is possible it could co-exist with LBRF and may further increase the health burden on the population, as well as a complicated diagnosis. Ticks from both countries were confirmed as *Ornithodoros savignyi*, an established vector for the above spirochaete. Ticks showed geographical clustering of sequence types based upon their country of origin, albeit all clearly belonged to the *O. savignyi* complex. We also detected the presence of *Borrelia theileri* in goats from a North-Central state in Nigeria. Available evidence shows that bovine borreliosis can pose huge veterinary and economic consequences. Given the recognition of *C*. B. kalaharica in humans with TBRF, investigation of the clinical impact of this spirochaete among humans exposed to the bite of *O. savignyi* is clearly a priority. 

## Figures and Tables

**Figure 1 pathogens-12-00081-f001:**
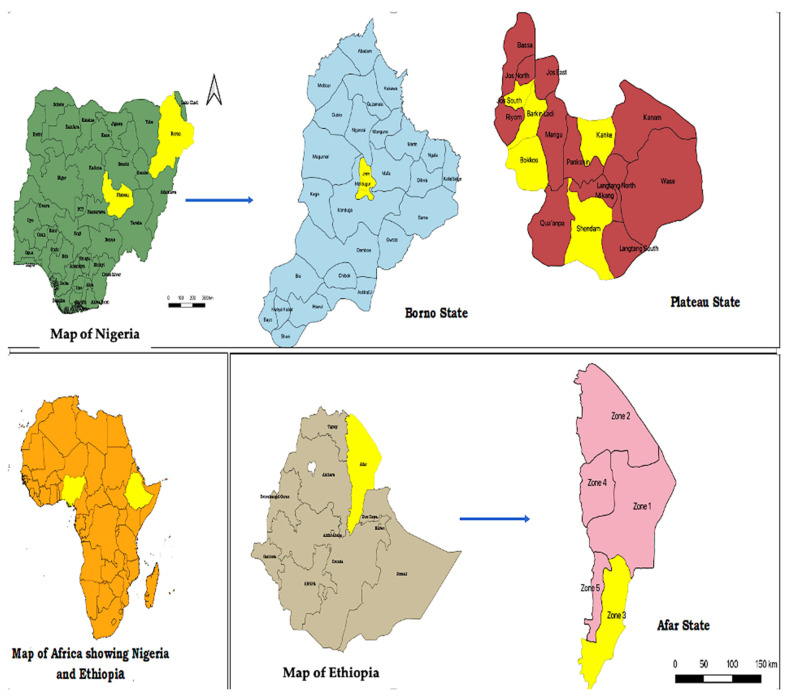
Map of Africa showing the study locations in Plateau (Shendam, Kanke, Barkin-Ladi, Jos South, Bokkos,) and Borno states (Maiduguri) Nigeria, and Ethiopia (Amibara) zone 3, Afar state. Maps were produced using QGIS version 3.10.

**Figure 2 pathogens-12-00081-f002:**
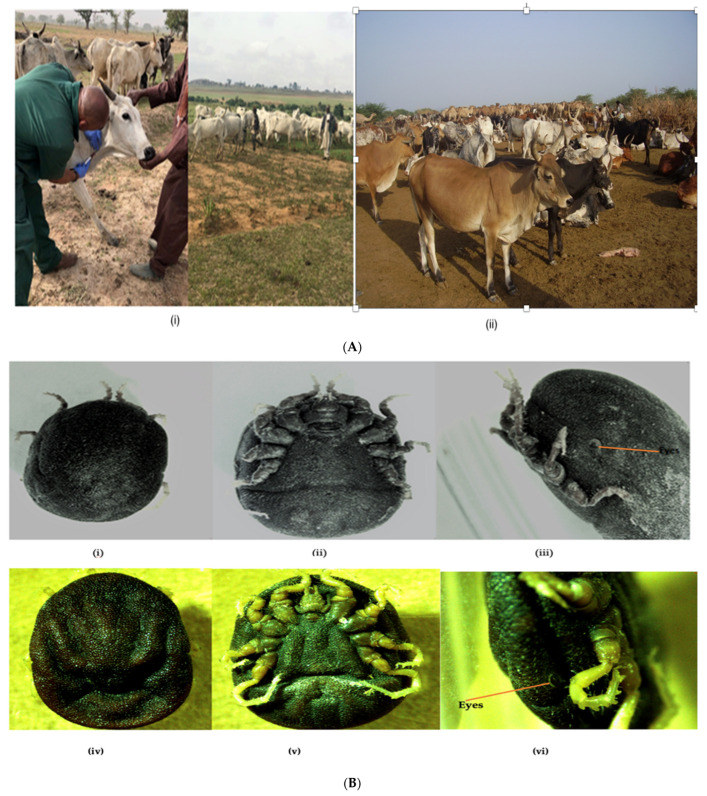
(**A**) (**i**) Livestock farms in Plateau and Borno states where animals were sampled; (**ii**) Livestock farm in Halidage village, Amibara district, zone 3, eastern Afar state where soft ticks were sampled. (**B**) Nigerian O. savignyi engorged adult (**i**) dorsal view (**ii**) ventral view (**iii**) non-functional eye. Ethiopian O. savignyi adult male (**iv**) dorsal view (**v**) ventral view (**vi**) non-functional eye.

**Figure 3 pathogens-12-00081-f003:**
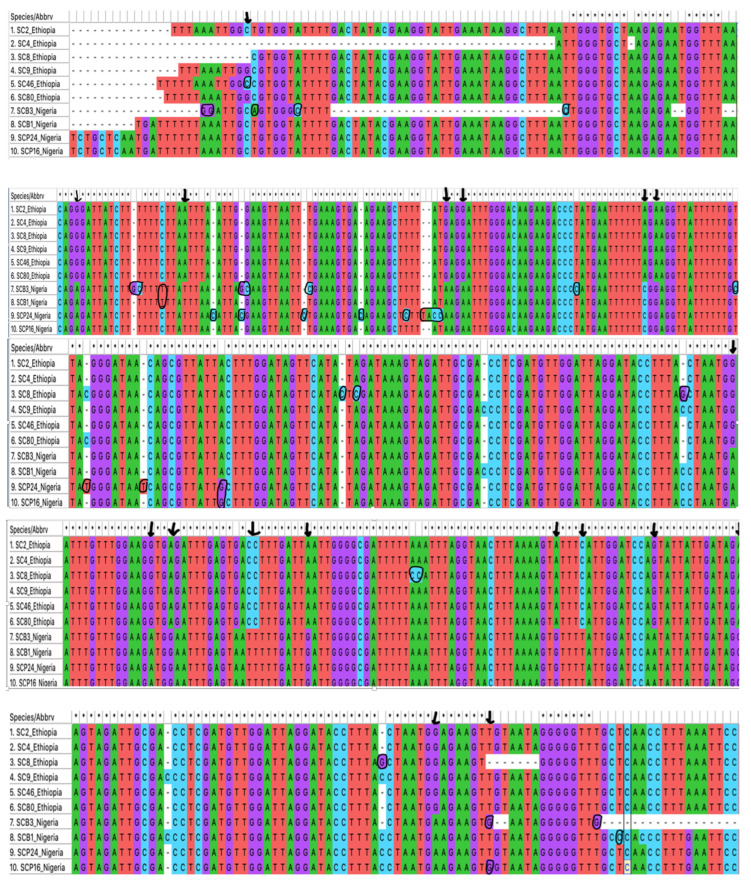
Differences in 16S rRNA (450 bp) sequences between the Ethiopian (numbers 1–6) and Nigerian (numbers 7–10) ticks, with the black arrows depicting location of the polymorphisms. Divergences among sequences from the same country of origin are circled.

**Figure 4 pathogens-12-00081-f004:**
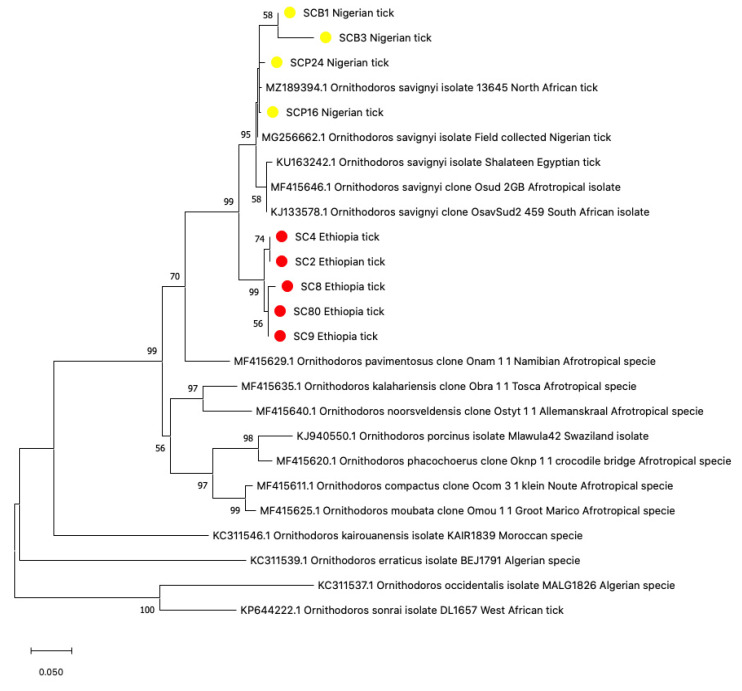
Phylogenetic analysis of Neighbor-joining tree of tick 16S rRNA sequences (450 bp), comparing the Nigerian and Ethiopian ticks with other *Ornithodoros* species. The tree with the highest log likelihood (−3939.77) is shown. Evolutionary distances were calculated using the maximum likelihood method in MEGA 11. Bootstrap values were >50% based on a test of confidence of 1000 replicates shown on branch nodes. The yellow and red circles represent the Nigerian and Ethiopian tick sequences respectively generated from this study.

**Figure 5 pathogens-12-00081-f005:**
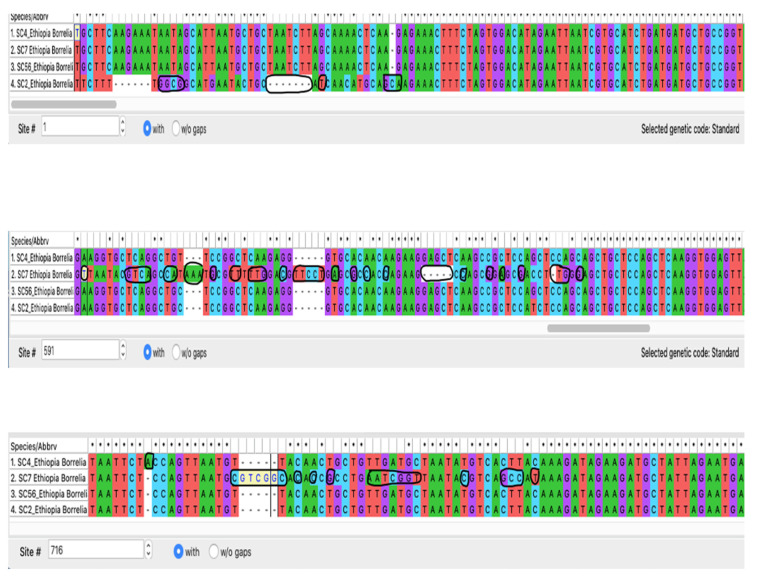
Divergences in the borrelial flagellin sequences at 770 bp among the Ethiopian ticks circled in black, with SC7 having the most polymorphisms.

**Figure 6 pathogens-12-00081-f006:**
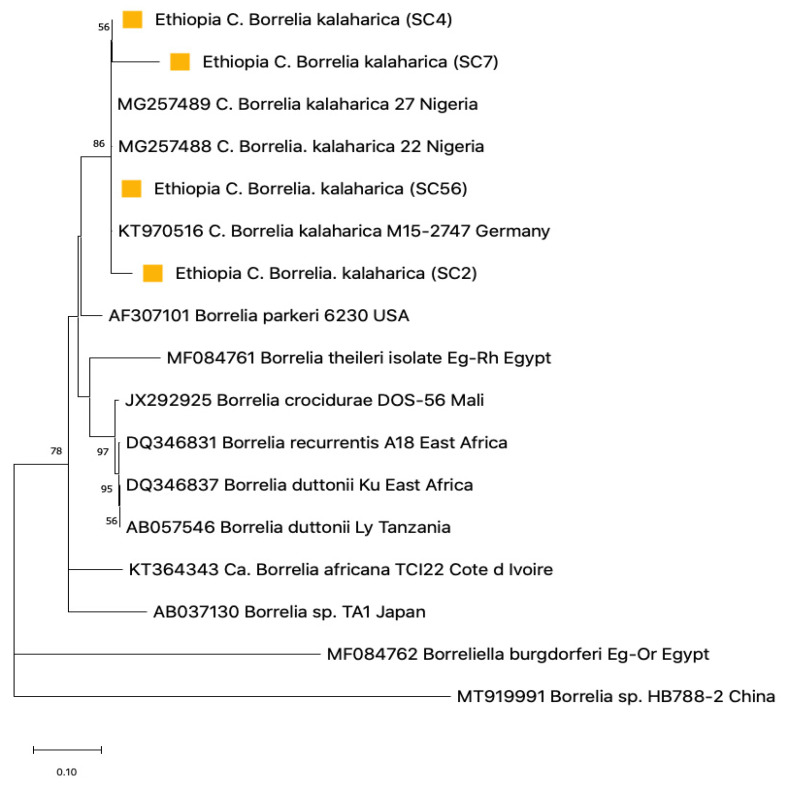
Phylogenetic analysis of Neighbor-joining of Borrelia flagellin B tick DNA sequences (770 bp) from Ethiopian ticks. The tree with the highest log likelihood (−3939.77) is shown. Evolutionary distances were calculated using the maximum likelihood method in MEGA 11. Bootstrap values were >50% based on a test of confidence of 1000 replicates shown on branch nodes. The squares signify the tick *Borrelia* sequences generated from this study.

**Figure 7 pathogens-12-00081-f007:**
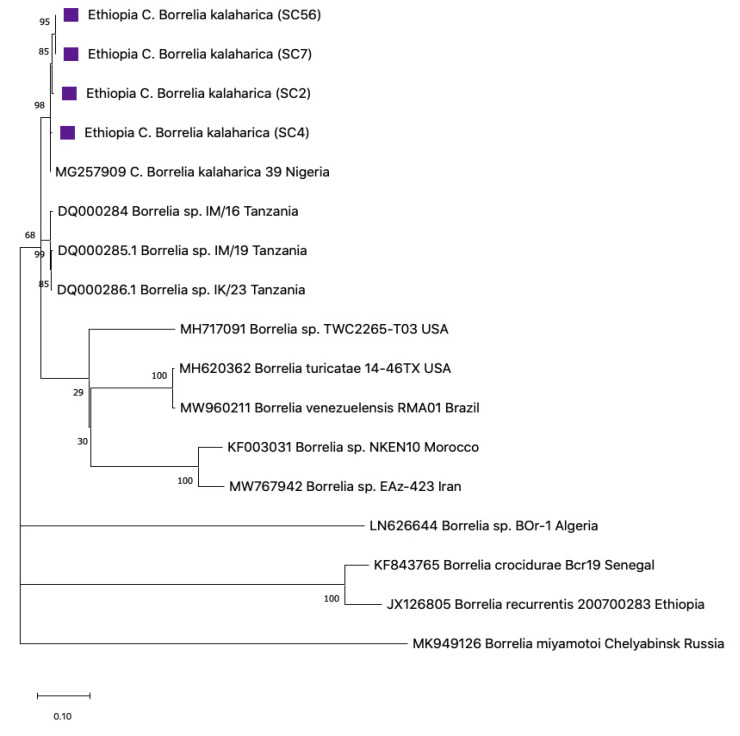
Phylogenetic analysis of Neighbor-joining of Borrelia IGS tick DNA sequences (750 bp) from Ethiopian ticks. The tree with the highest log likelihood (−5105.81) is shown. Evolutionary distances were calculated using the maximum likelihood method in MEGA 11. Bootstrap values were >50% based on a test of confidence of 1000 replicates shown on branch nodes. The squares signify the tick *Borrelia* sequences generated from this study.

**Figure 8 pathogens-12-00081-f008:**
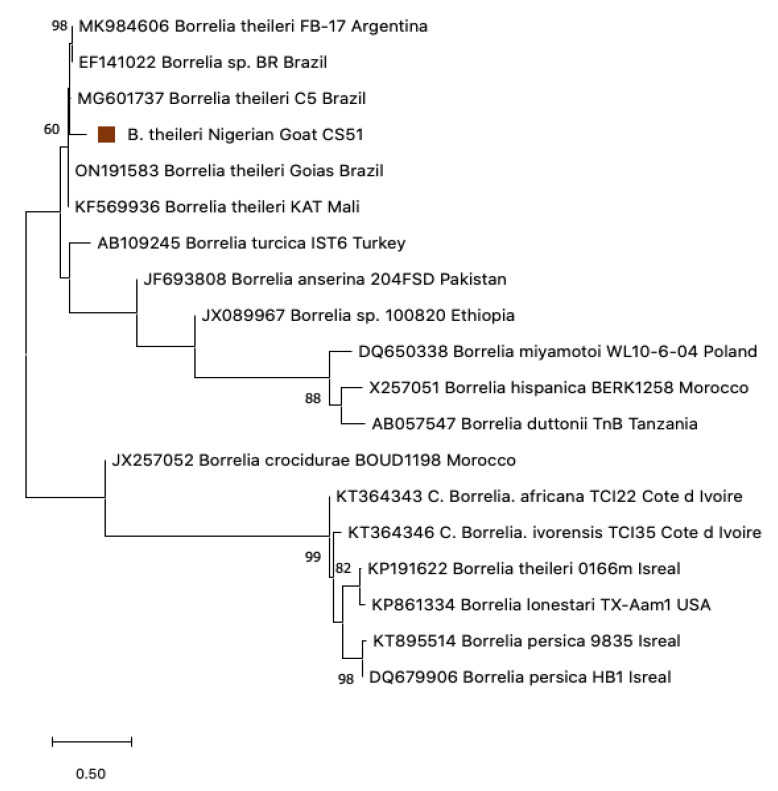
Phylogenetic analysis of Neighbor-joining of Borrelia flagellin goat DNA sequences (350 bp) from Nigeria. The tree with the highest log likelihood (−2182.93) is shown. Evolutionary distances were calculated using the maximum likelihood method and Tamura 3 models in MEGA 11. Bootstrap values were >50% based on a test of confidence of 1000 replicates shown on branch nodes. The square signifies the goat *Borrelia* sequence generated from this study.

**Figure 9 pathogens-12-00081-f009:**
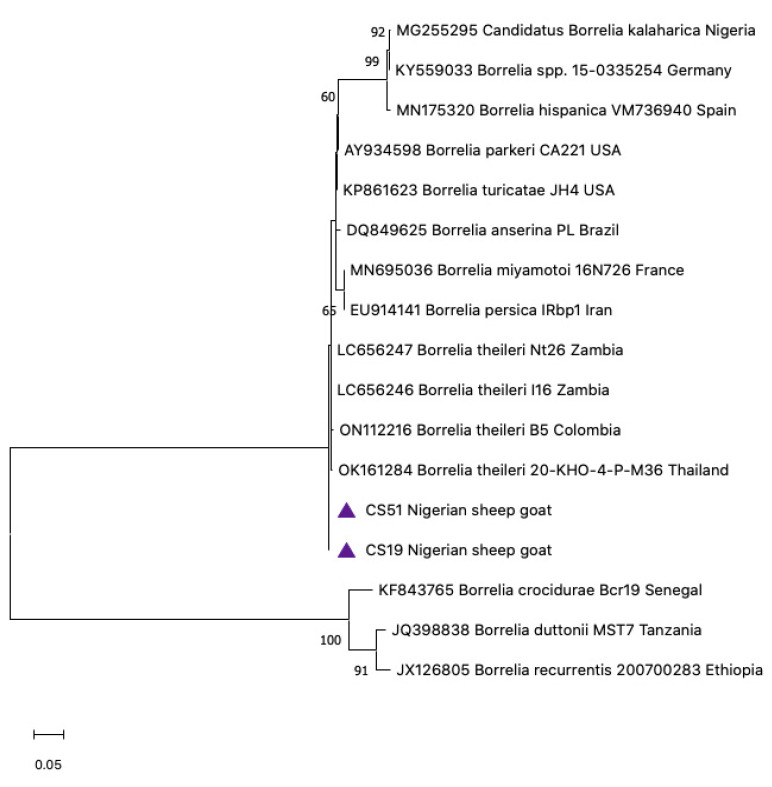
Phylogenetic analysis of Neighbor-joining of *Borrelia* 16S rRNA from goat sera (650 bp) from Nigeria. The tree with the highest log likelihood (−133.32) is shown. Evolutionary distances were calculated using the maximum likelihood method and Tamura 3 models in MEGA 11. Bootstrap values were >50% based on a test of confidence of 1000 replicates shown on branch nodes. The triangles represents the goat *Borrelia* sequences generated from this study.

**Table 1 pathogens-12-00081-t001:** Oligonucleotides used for this study.

Genome Target	Nucleotide Sequences	Band Size	Reference
16S–23S intergenic spacer region (IGS) (Ethiopia soft ticks)	First round: F: GTATGTTTAGTGAGGGGGGTG R: GGATCATAGCTCAGGTGGTTAG Nested: F: AGGGGGGTGAAGTCGTAACAAG R: GTCTGATAAACCTGAGGTCGGA	750 bp	[17]
Flagellin B (flaB) (Ethiopia soft ticks)	F: TAATACGTCAGCCATAAATGC R: GCTCTTTGATCAGTTATCATTC	770 bp	[33]
Flagellin B (flaB) (Nigerian livestock)	First round: F: GATCARGCWCAAYATAACCAWATGCA R: AGATTCAAGTCTGTTTTGGAAAGC Nested: F: GCTGAAGAGCTTGGAATGCAACC R: TGATCAGTTATCATTCTAATAGCA	350 bp	[34]
16S rRNA (Nigerian livestock)	First round: F: GCGAACGGGTGAGTAACG R: CCTCCCTTACGGGTTAGAA Nested: F: GAGGCGAAGGCGAACTTCTG R: CTAGCGATTCCAACTTCATGAAG	650 bp	[35]
RT-PCR 16S rRNA (All samples)	F: AGCCTTTAAAGCTTCGCTTGTAG R: GCCTCCCGTAGGAGTCTGG P [**FAM**] CCGGCCTGAGAGGGTGAACGG	148 bp	[36]
Tick 16S rDNA (All tick samples)	F: CTGCTCAATGATTTTTTAAATTGC R: CCGGTCTGAACTCAGATCATGTA	450 bp	[37]

**Table 2 pathogens-12-00081-t002:** PCR result of all samples.

Species	RT-PCR	Flagellin B	16S–23S IGS	16S rRNA
*O. savignyi* (Ethiopia)	3.5% (11/312)	4/11	4/11	NA
*O. savignyi* (Nigeria)	0/77	NA	NA	NA
Cattle	5% (18/350)	0/18	0/18	0/18
Goat	1.5% (3/200)	3/3	3/3	3/3

NA: Not applicable.

**Table 3 pathogens-12-00081-t003:** Prevalence of *Borrelia* in Ethiopian ticks of all developmental stages.

Stage/State	16S RT-PCR			Flagellin	IGS	
	*Borrelia* infection vs. total ticks screened	*Borrelia* positive (engorged ticks)	*Borrelia* positive (non-engorged ticks)	Amplified *Borrelia* sequences	Amplified *Borrelia* sequences	*p*-value
Adult ticks	10% (6/60)	36% (4/11)	18% (2/11)	27% (3/11)	27% (3/11)	*p* * > 0.25
Nymphs	2% (5/252)	27% (3/11)	18% (2/11)	9% (1/11)	9% (1/11)	*p* ** > 0.51
Total screened	11/312	7/11	4/11	36% (4/11)	36% (4/11)	

*p* * prevalence of *Borrelia* infection between adults and nymphs after confirmation on flaB and IGS. *p* ** prevalence of *Borrelia* infection between engorged and non-engorged ticks after confirmation.

## Data Availability

The data that supports the findings of this study are available from the lead author upon reasonable request.

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
