# Peer review of "Comparative Analysis of Tick-Borne Relapsing Fever Spirochaetes from Ethiopia and Nigeria"

_pathogens, 2023, doi:10.3390/pathogens12010081_

Round 1
Reviewer 1 Report
Excellent manuscript. The information provided in this manuscript will be useful when designing Relapsing Fever tests for Africa.
Reviewer 2 Report
General comments:
Language editing would substantially improve quality of the manuscript, some parts of the text are difficult to understand. I’m suggesting that senior author as native speaker perform editing of the manuscript.
General focus of the manuscript is lacking. What was the main idea of the authors to present presence and variability of the pathogen or both pathogen and vector? Title of the manuscript is not informative enough, it’s rather general and I suggest to adjust it after deciding on the main focus of the manuscript.
Material and Methods
I suggest to use map of Africa to illustrate the location and distance of Ethiopia and Nigeria, and investigated localities. Authors could corelate such figure with observed differences in genetic structure of populations of soft ticks.
L122 to L125 – Either describe complete methodology for DNA extraction or cite reference if already published protocol was followed. From the sentence written in this part exact methodology is not clear.
L144 to L147 – Provide cycling conditions for conventional PCR protocols.
Table 1. Exclude shadings from the table, it’s difficult to follow the information. Format table.
Results
L194 to L197 Did authors perform only molecular identification or morphological as well? Successful rate of molecular identification is quite low, the reason could be that protocol is not adjusted or extraction of DNA was not successful for all samples.
Figure 3. Provide higher quality figure, this one is blurred. Authors should provide row chromatograms for sequences included in the presented analysis in this table and in other phylogenetic analysis, as supplementary material.
L237 to L239. The authors refer to tick life stages and engorgement without previous information on morphological examination on ticks, and number of ticks that belong to each life stage. Results are partially presented in Table 3. however, it not clear based on what authors calculate presented prevalence for engorged and unengorged ticks.
Discussion
L367 to L368 – Provide information on the discriminatory limit between species for 16S rDNA gene. The identity of 93-94% is quite low, is it enough for identification of samples as Ornithodoros savignyi?
L386 to L388. Based on what identification method authors state that analyzed ticks belong to Ornithodoros savignyispecies? Number of molecularly identified samples is quite low and can’t be extrapolated on all tick samples.
L390 to L391. If both Ixodes ricinus and Ornithodoros savignyi positive controls were successfully amplified with protocol used for molecular identification of ticks it’s not likely that failure to amplify more tick samples is due to sequence heterogeneity among Ornithodoros savignyi samples.
Reviewer 3 Report
I have carefully reviewed the paper on Expanding Diversity of Relapsing fever Spirochaetes in African Soft Ticks that you recommended for consideration, and my comments are as follows.
1. Tick-borne diseases bring great suffering to the life and health of African people. It is very necessary to monitor tick-borne diseases and find diagnosis and treatment measures, especially for soft ticks in the African continent, there are few studies. This topic is correct and very meaningful.
2. The overall structure of this paper is not a big problem, the idea is clear, the method is feasible, it is recommended to be included, but to do the following modifications.
3. Most of the figures and typography of this submitted paper is of poor quality and must be replaced or modified.
4. The submitted paper explains that the pathogen can be transmitted by birds, that the distance between the two countries is thousands of kilometers, that trade is introduced, but it does not mention the role of animal migration, nor does it mention why Nigerian ticks are more similar to avian ticks than Ethiopian ticks. The paper does not explain.
5. In this paper, there are too few tick monitoring sites, and the sample size is too small. If the monitoring area can be expanded and more samples can be obtained, better results may be achieved.
6. The paper notes that spirochete was not detected in Nigerian tick samples, but tested positive for borrelia in domestic animals. Personally, I think it is related to the small number of ticks, in addition, it is also related to the collection and preservation time and different hosts, which should be analyzed from a larger scale in the paper.
As I do not know much about African soft ticks and tick-borne diseases, I cannot offer more profound insights. The above views are for reference only.
Round 2
Reviewer 2 Report
I wish to thank the authors for addressing all my comments and suggestions in the appropriate manner providing detailed answers and significantly improving the quality of the paper. I found it suitable for publication in its present form.
